# Critical Reappraisal of Methods for Measuring Urine Saturation with Calcium Salts

**DOI:** 10.3390/molecules26113149

**Published:** 2021-05-25

**Authors:** Silvia Berto, Martino Marangella, Concetta De Stefano, Demetrio Milea, Pier Giuseppe Daniele

**Affiliations:** 1Dipartimento di Chimica, Università di Torino, via P. Giuria 7, 10125 Torino, Italy; pierodan2015@gmail.com; 2Fondazione Scientifica Mauriziana-Onlus, via Magellano 1, 10128 Torino, Italy; mmarangella@alice.it; 3Dipartimento di Scienze Chimiche, Biologiche, Farmaceutiche ed Ambientali, Università degli Studi di Messina, CHIBIOFARAM, Viale Ferdinando Stagno d’Alcontres 31, 98166 Messina, Italy; cdestefano@unime.it (C.D.S.); dmilea@unime.it (D.M.)

**Keywords:** nephrolithiasis, calcium oxalate, calcium phosphate, citrate, chemical speciation, oversaturation

## Abstract

*Background*: Metabolic and physicochemical evaluation is recommended to manage the condition of patients with nephrolithiasis. The estimation of the saturation state (β values) is often included in the diagnostic work-up, and it is preferably performed through calculations. The free concentrations of constituent ions are estimated by considering the main ionic soluble complexes. It is contended that this approach is liable to an overestimation of β values because some complexes may be overlooked. A recent report found that β values could be significantly lowered upon the addition of new and so far neglected complexes, [Ca(PO_4_)Cit]^4−^ and [Ca_2_H_2_(PO_4_)_2_]. The aim of this work was to assess whether these complexes can be relevant to explaining the chemistry of urine. *Methods:* The Ca–phosphate–citrate aqueous system was investigated by potentiometric titrations. The stability constants of the parent binary complexes [Cacit]^−^ and [CaPO_4_]^−^, and the coordination tendency of PO_4_^3−^ toward [Ca(cit)]^−^ to form the ternary complex, were estimated. β_CaOx_ and β_CaHPO4_ were then calculated on 5 natural urines by chemical models, including or not including the [CaPO_4_]^−^ and [Ca(PO_4_)cit]^4−^ species. *Results:* Species distribution diagrams show that the [Ca(PO_4_)cit]^4−^ species was only noticeable at pH > 8.5 and below 10% of the total calcium. β values estimated on natural urine were slightly lowered by the formation of [CaPO_4_]^−^ species, whereas [Ca(PO_4_)cit]^4−^ results were irrelevant. *Conclusions:* While [CaPO_4_]^−^ species have an impact on saturation levels at higher pHs, the existence of ternary complex and of the dimer is rejected.

## 1. Introduction

Nephrolithiasis (NL) is a common disease that is characterized by high rates of recurrence [1]. Medical management is aimed at preventing stone reappearance. It is generally agreed that the metabolic and physicochemical evaluation of urine may help to guide physicians to choose adequate therapies and to control for patients’ compliance [2]. Concerning calcium NL, the risk of forming stones stems from an imbalance between urine components promoting (e.g., calcium, oxalate, and phosphate) and inhibiting (e.g., magnesium, citrate, and macromolecules) NL [3]. The interactions among these moieties regulate crystallization through thermodynamic and/or kinetic processes. Whereas techniques for measuring crystal kinetics are mainly performed for research purposes only, being time-consuming and poorly standardized, the thermodynamic approach, based on the assessment of the urine saturation state, is often included in the metabolic evaluation of patients. There are essentially two methods to perform such measurements: the semi-empirical one proposed by Pak et al. [4], and the computer-based calculations formerly proposed by Robertson et al. [5]. While Pak’s method has been seldom used due to its complexity, several computer-based calculations, including EQUIL [6], JESS [7], and Lithorisk^®^ [8], have been proposed and recommended in the management of patients with NL [9].

While such calculations represent a valuable approach to estimating the tendency to form stones, some problems remain concerning the comparability between methods. In principle, calculations are based on the estimation of the free ion concentration product of the constituent ions of calcium salts. This can be provided if the most important soluble complexes of calcium, oxalate and phosphate are introduced into the calculation [10]. The complexes formed depend on the components of urine, and the concentration of the complexes depends on their stability constant. The components of the speciation models were selected based on the Geigy tables [11]. Different results between methods may arise from differences in the number and type of stability constants used. The JESS (joint expert speciation system) was recently developed, based on the availability of an extensive database of thermodynamic constants [12]. When compared to the previous EQUIL [10] method, it was shown to yield significantly lower estimations of relative supersaturation with both calcium oxalate (calcium oxalate monohydrate, COM) and calcium phosphate (brushite, BSH). This was accounted for by JESS considering two so far neglected complex species, namely, [Ca(PO_4_)Cit]^4−^ and [Ca_2_H_2_(PO_4_)_2_]. In a former paper we reported a fair correspondence between our Lithorisk^®^ method [8] and EQUIL [10]. Therefore, similar differences are expected to emerge by comparing JESS to Lithorisk^®^. In this paper, we aim to evaluate whether the aforementioned complexes can form in common urinary conditions and be consequently relevant in measurements of urine saturation. In order to evaluate whether these complexes form in urine on the basis of thermodynamics, the Ca–phosphate–citrate aqueous system was studied by pH- and pCa-metric titrations, at an ionic strength of *I* = 0.1 mol L^−1^ (using KCl as the ionic medium) and *t* = 37 °C. Since an accurate evaluation of the formation constant of a ternary mixed complex requires the knowledge of the stability constants of the parent binary complexes [Cacit]^−^ [13] and [CaPO_4_]^−^, we concurrently studied the Ca phosphate system in the same experimental conditions. The pH range considered for both of the above systems was 5.0–7.5, where the dissociation of the last proton of phosphate anion can occur.

## 2. Results and Discussion

### 2.1. Stability of Calcium Mixed Complexes

Alkalimetric titrations were conducted on the Ca–phosphate and Ca–phosphate–citrate aqueous systems until the abrupt variation of potentials occurred, revealing the beginning of the precipitation of calcium-phosphate salts. An example of titration curves obtained on the Ca–phosphate–citrate system, recorded by both ISE-H^+^ and ISE-Ca^2+^ (ion-selective electrode (ISE)), is shown in Figure 1.

The pH-metric data collected on the Ca–phosphate aqueous system were used to determine the formation constant of the [CaPO_4_]^–^ species. Concerning the dimeric [Ca_2_H_2_(PO_4_)_2_] species, we had already reported no evidence of it when determining the stability constant of the monomeric [CaHPO_4_] species [14], and this was confirmed by results obtained in our experimental conditions. In fact, the experimental data cannot be explained by chemical models containing the dimeric species. The inclusion of this species in models has always led to a marked discrepancy between the experimental curves and those calculated through the application of the model.

The optimized value of [CaPO_4_]^–^ species is log*K* = 6.15 ± 0.02 (Table 1), which agrees with the results reported by Nancollas et al. [15]: log*K* = 6.13. The pCa values obtained by the ISE-Ca^2+^ measurements agree with those calculated by the chemical model (for further details on data analysis, see: Section 4.4).

By taking into account the formation constants of both parent binary complexes [Cacit]^–^ and [CaPO_4_]^−^, pH-metric data collected by titrating ternary mixtures containing Ca(II), citrate and phosphate, were analyzed with the aim of evidencing the formation of ternary species. Of course, only the potentiometric data registered before the precipitation of calcium phosphates were considered. The value of the formation constant calculated for the [Ca(PO_4_)cit]^4−^ complex is logβ = 8.3 ± 0.8. The relatively high uncertainty estimated is due to a negligible formation percentage (Table 1).

In order to characterize the coordination tendency of the ligand PO_4_^3−^ toward [Ca(cit)]^−^, the value of Δlog*K*_Ca_ was calculated, as suggested by H. Sigel [16]. Δlog*K*_Ca_ is the difference between the partial formation constant of the mixed complex—$\log K_{{\mathrm{CacitPO}}_{4}}^{\mathrm{Cacit}}$ and the partial formation constant of the complementary complex—${\log K}_{{\mathrm{CaPO}}_{4}}$:(1)$$\log K_{{\mathrm{CacitPO}}_{4}}^{\mathrm{Cacit}}={\log\beta }_{{\mathrm{CacitPO}}_{4}}-\log K_{\mathrm{Cacit}}$$
(2)$$\Delta \log K_{\mathrm{Ca}}=\log K_{{\mathrm{CacitPO}}_{4}}^{\mathrm{Cacit}}-\log K_{{\mathrm{CaPO}}_{4}}$$

Since the log*K* for [CaPO_4_]^−^ is 6.15, the Δlog*K*_Ca_ is −1.48. The Δlog*K*_Ca_ quantifies the stability of the ternary complex, relative to the binary parent complex. The statistically expected value of Δlog*K*_Ca_, evaluated by H. Sigel [16] for the coordination of two different bidentate ligands to a regular octahedral coordination sphere, is −0.38. The value of Δlog*K*_Ca_ obtained for [Ca(PO_4_)cit]^4−^ is more strongly negative than that statistically expected. This outcome suggests that the formation of the ternary complex is quite improbable, and this is probably due to a charge repulsion factor. The negative charges of both the parent complexes and the second ligand hinder the formation of the mixed complex.

The possibility to explain the titration data by only considering the [CaPO_4_]^−^ species on the one hand, and the high uncertainty of [Ca(PO_4_)cit]^4−^ formation constants on the other, reinforce the hypothesis that the formation of the mixed species is quite unlikely.

Figure 1 shows the simulated titration curves derived from the application of the speciation model reported in Table 1, and Figure 2 illustrates the speciation diagrams for the solution under study. The figures show that the percentage of Ca^2+^ involved in the [Ca(PO_4_)cit]^4−^ species is negligible. Only at a pH higher than 8.5, which is uncommon in urine, the formation percentage of [Ca(PO_4_)cit]^4−^ becomes noticeable, though it is always below 10% of total calcium. On the other hand, with pH as high as this, the struvite ((NH_4_)MgPO_4_×6(H_2_O)) stone formation becomes by far the most clinically relevant.

### 2.2. Calcium Complexes in Urinary Conditions

The relevance of the species here defined, [CaPO_4_]^−^ and [Ca(PO_4_)cit]^4−^, on the saturation conditions of real urine samples (the pH and the molar concentrations of the urine components are reported in Appendix A) was tested. A chemical model that involves the species reported in Appendix A (hereinafter Model 2) was applied to 5 natural urine samples with a pH between 6.5 and 7.5. The results were compared with those obtained by applying the previous model (hereinafter Model 1), which ignores the [CaPO_4_]^−^ and [Ca(PO_4_)cit]^4−^ species. The formation constants of the herein considered species are the same as those previously used in speciation studies of some biological fluids [21,22]. On each sample, the species distribution and the saturation level with the two models were defined. The saturation level was represented by the values of β_CaOx_ and β_CaHPO4_ calculated as previously described [23] and summarized in the Appendix A.

The results of the saturation levels for each urine sample, obtained by applying the two chemical models, are reported in Table 2. Figure 3 shows the species distribution diagrams of one sample of urine (the other distribution plots were collected in Appendix A). The species distribution diagrams were plotted in the pH range of 4.0–8.5, even if a pH higher than 7.5 is very unusual for natural urines, unless infected with urease-producing bacteria. Such a wide pH range was used to stress the formation of the [CaPO_4_]^−^ and [Ca(PO_4_)cit]^4−^ species. The speciation diagrams show that, while the [CaPO_4_]^−^ species starts to be relevant at pH > 7, the mixed [Ca(PO_4_)cit]^4−^ species appears to be completely negligible in this urine sample. Therefore, the changes in β_CaOx_ and β_CaHPO4_ values obtained through Model 2, compared to Model 1, are only due to the addition of [CaPO_4_]^−^ species in the former.

In order to verify how an increase in citrate concentration affects the species distribution and the saturation values, the speciation diagram of urine B, modified by increasing total citrate concentration to 4.0 × 10^−3^ mol L^−1^, was drawn as shown in Figure 3e. This test was performed on urine B because this is a sample with particularly high pH and Ca^2+^ concentration, which could enhance the [Ca(PO_4_)cit]^4−^ formation (see Appendix A). Under these conditions, [Ca(cit)]^−^ becomes the main species in the pH range 5–8, thereby reducing the percentage of free calcium and [Ca(ox)]. Estimating the saturation values of urine B, it is possible to note a strong decrease of β values upon increasing the citrate concentration: going from a citrate concentration of 8.5 × 10^−4^ mol L^−1^ to 4.0 × 10^−3^ mol L^−1^, the pH shifts from 7.24 to 7.27, and the β_CaOx_ and β_CaHPO4_ change from 5.74 and 4.73 to 3.69 and 2.93, respectively (Table 2). The decrease in saturation values cannot be explained by the presence of mixed species. [Ca(PO_4_)cit]^4−^ remains absent up to pH 7 and reaches a percentage of only 13% at pH 8.5.

## 3. Discussion

The chemical model previously proposed for measuring urine saturation was now reappraised. The method is based on the assessment of the saturation level through the estimation of free concentrations of calcium, oxalate, and hydrogen phosphate. The free concentrations were estimated by the application of a chemical model that comprises all chemical equilibria that can affect the formation of calcium oxalate and calcium phosphate species. An analogous approach was used for speciation studies of different biological fluids, providing reliable results [21], and this may be applied to other biological fields.

Previous reports [12] suggested that a citrate-related increase in urine pH might result in an additional decrease in saturation levels, due to an augmented formation of the [CacitPO_4_]^4−^ complex. The results reported in this work suggest, instead, that the strong decrease in urine saturation with calcium salts obtained upon citrate addition was mainly due to [Ca(cit)]^−^ and not to [CacitPO_4_]^4−^. In other words, the favorable effect of citrate supplementation to calcium stone-forming patients should be ascribed to the consequent enhancement of urinary citrate rather than to an increase in urine pH. Furthermore, because higher urine pH increases the risk of calcium phosphate, clinicians should carefully manage citrate supplementation, paying attention to its excessive enhancement in urine pH.

## 4. Materials and Methods

### 4.1. Chemicals

Sodium phosphate dibasic, purity ≥ 99.5%, was sourced from Merck (Darmstadt, Germany). Calcium nitrate tetrahydrate, purity 99%, was sourced from Sigma Aldrich (Darmstadt, Germany). Potassium chloride, purity ≥ 99%, and sodium citrate dibasic sesquihydrate, purity ≥ 99%, were sourced from Carlo Erba Reagents (Cornaredo, Italy).

Standard KOH 0.1 mol L^−1^ and HCl 0.1 mol L^−1^ solutions were prepared by diluting Merck (Darmstadt, Germany) concentrate products. The KOH solution was standardized against potassium hydrogen phthalate (Fluka, Puriss; Darmstadt, Germany). All solutions were prepared using grade A glassware and ultrapure water (conductivity < 0.1 μS).

A calcium stock solution (0.1 mol L^−1^) was prepared by dissolving calcium nitrate tetrahydrate in ultrapure water and titrated by EDTA.

### 4.2. Apparatuses

Potentiometric measurements were performed using the Metrohm mod. 713 potentiometer (resolution of ±0.1 mV) coupled with a Metrohm combined glass electrode (mod. 6.0259.100), with internal reference Ag/AgCl/3M KCl, and the Amel mod. 338 potentiometer (resolution of ±0.1 mV) coupled with a Radiometer Analytical calcium selective electrode (mod. ISE25Ca) and a Metrohm reference electrode Ag/AgCl/3M KCl (mod. 6.0733.100). A Metrohm 765 Dosimat burette (minimum volume deliverable of ±0.001 cm^3^) was used to deliver the titrant solution. The potentiometric titrations were carried out in a stream of purified nitrogen gently bubbled in the titration cell to avoid O_2_ and CO_2_ contamination. The measurement cells are maintained at a constant temperature of 37 ± 0.1 °C by means of water circulation from a thermocryostat (mod. D1-G Haake).

### 4.3. Calibration and Titration Procedures

The glass electrode was calibrated, in terms of pH = −log[H^+^], by titrating HCl 5 mmol L^−1^ solution at the same ionic strength value as the solution under study, with standard KOH, in order to determine the formal potential *E*^0^ before each experiment. The electrode calibration data were analyzed by the ESAB2M program [24] to refine the electrode parameters. This program was used to refine the formal potential *E*^0^, the Nernstian slope at 37 °C, and the analytical concentration of the reagents.

The ISE-Ca^2+^ was calibrated in −log[Ca^2+^] units (pCa), recording the emf (electromotive force, mV) values obtained by adding known volumes of calcium stock solution 0.1 mol L^−1^. The Ca^2+^ concentration of the calibrating solutions ranged between 2 × 10^−4^ and 5 × 10^−3^ mol L^−1^. The recorded values of emf were used for the internal calibration and allowed to calculate the slope and the *E*^0^ of the Nernst equation for the ISE-Ca^2+^ at 37 °C. The procedure was applied before the addition of the ligands, citrate and phosphate. Then, alkalimetric titrations were carried out. The concentrations of the calcium in solutions ranged between 2 × 10^−3^ and 5 × 10^−3^ mol L^−1^, and the Ca^2+^ to citrate ratio used was 1:1, while the phosphate was always in excess with respect to the cation, and the phosphate to Ca^2+^ ratios were maintained between 1.5 and 4. The concentration ratios between the components of the solutions were chosen based on the concentrations of these ions in the urine samples. The ionic strength was adjusted to 0.1 mol L^−1^ with KCl. The solutions were titrated with KOH standard solution 0.1 mol L^−1^, and both pH and pCa were monitored during the titration. The pH range considered was 5.0–7.5. The titrations were carried out until the precipitation of calcium phosphate salt occurred. Each titration was repeated at least twice.

### 4.4. Data Treatment

The pH-metric titration data were analyzed using BSTAC software [25] to determine the formation constants. The software employs an iterative and convergent numerical method, which is based upon the linear combination of the mass balance equations, minimizes the error squares sum on emf values, and considers possible variations of ionic strength among and/or during titrations. The software allows for optimizing the concentration of the components of the solution under study, and possibly constraining them.

The pCa values obtained by the measurements with ISE-Ca^2+^ were not used to define the formation constants of the Ca^2+^ because the overall variation of potential recorded by the ISE-Ca^2+^ during the titration process was very low, ~5 mV, and this technique was considered to have unsuitable sensitivity. However, the experimental pCa values were compared with those calculated by the application of the chemical model to verify the consistency between them.

The species considered in the chemical model used for the data elaboration are reported in Table 1. The formation constants of the species were derived from literature data when possible (see refs. reported in Table 1). Suitable calculation processes were applied on the original literature values, if necessary, to obtain the log*K* values at the working conditions of ionic strength *I* = 0.1 mol L^−1^, KCl and *t* = 37 °C. In Table 1, both original and extrapolated values were reported. The effect due to ionic strength was considered by the application of an Extended Debye–Hückel (EDH) equation [17], whereas the effect of the temperature was evaluated by linear interpolation of the log*K* values from data collected at 20, 30 and 40 °C, respectively [18].

The weak interactions between the phosphate and citrate anions with the potassium cation were considered in the chemical model, therefore the protonation constants of the phosphoric and citric acids are those estimated with non-interacting cations [18]. The species distribution diagrams were drawn using HySS software [26].

## 5. Conclusions

This work was aimed at assessing experimentally whether the formation of the [CaPO_4_]^−^ and [Ca(PO_4_)cit]^4−^ species could be relevant in the chemical model proposed to explain the chemistry of urine, and consequently, to evaluate their effects on the saturation levels in urines at higher pH values. The formation constants of the two species were defined, and the new chemical model was tested on five different natural urine samples. Indeed, the [CaPO_4_]^−^ species was found to be of some relevance. The model in which it was included yielded slightly lower saturation values for all the tested urines, especially those with a higher pH. On the contrary, the [CacitPO_4_]^4−^ species results to be poorly significant in both synthetic and natural samples. At a urine pH of 7.5, the fraction of calcium included in this ternary complex remained negligible and was always below 5%.

The overall results of the present study make us confident to conclude that the existence of the ternary complex can be rejected. Conversely, the [CaPO_4_]^−^ species may have some impact on saturation levels, especially at higher pH, and should be taken into account for calculations. Therefore, this species will be included in the chemical model used by Lithorisk^®^ software [8] to take into account its formation in the estimation of saturation values, and to better assess the overall risk of kidney stones.

## Figures and Tables

**Figure 1 molecules-26-03149-f001:**
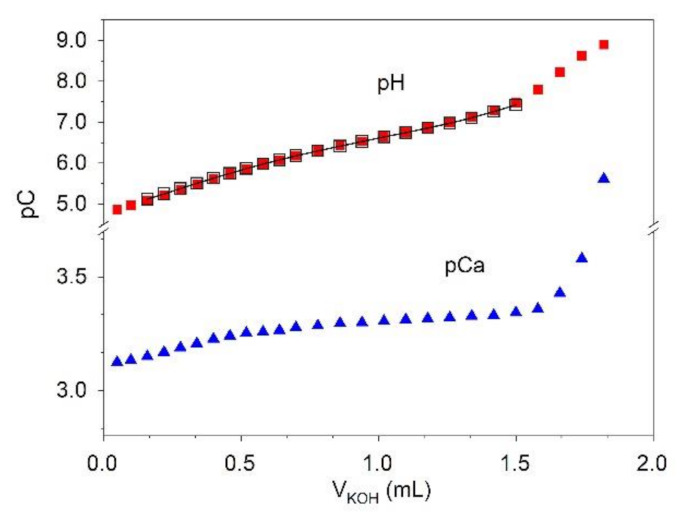
Titration curves recorded by both ISE-H^+^ and ISE-Ca^2+^ on a solution of Ca^2+^: 0.98 × 10^−3^ mol L^−1^; K^+^: 1.00 × 10^−1^ mol L^−1^; Cl^−^: 1.00 × 10^−1^ mol L^−1^; PO_4_^3−^: 2.35 × 10^−3^ mol L^−1^; citrate: 1.00 × 10^−3^ mol L^−1^; titrant: KOH 0.1 mol L^−1^ (*I* = 0.1 mol L^−1^; *t* = 37 °C). Full symbols: experimental points; empty black symbols—line: calculated values based on the speciation model reported in Table 1.

**Figure 2 molecules-26-03149-f002:**
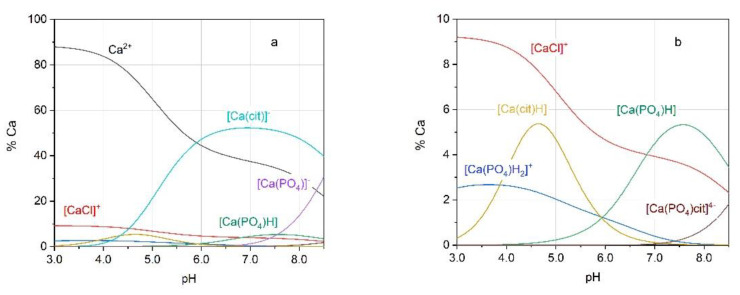
Speciation plots computed using the speciation model reported in Table 1. (**a**) Full figure; (**b**) zoomed figure showing the species at low formation percentage. The concentrations of the components are: Ca^2+^: 1.0 × 10^−3^ mol L^−1^; K^+^: 1.0 × 10^−1^ mol L^−1^; Cl^−^: 1.0 × 10^−1^ mol L^−1^; PO_4_^3−^: 2.5 × 10^−3^ mol L^−1^; citrate (cit): 1.0 × 10^−3^ mol L^−1^ (*I* = 0.1 mol L^−1^; *t* = 37 °C).

**Figure 3 molecules-26-03149-f003:**
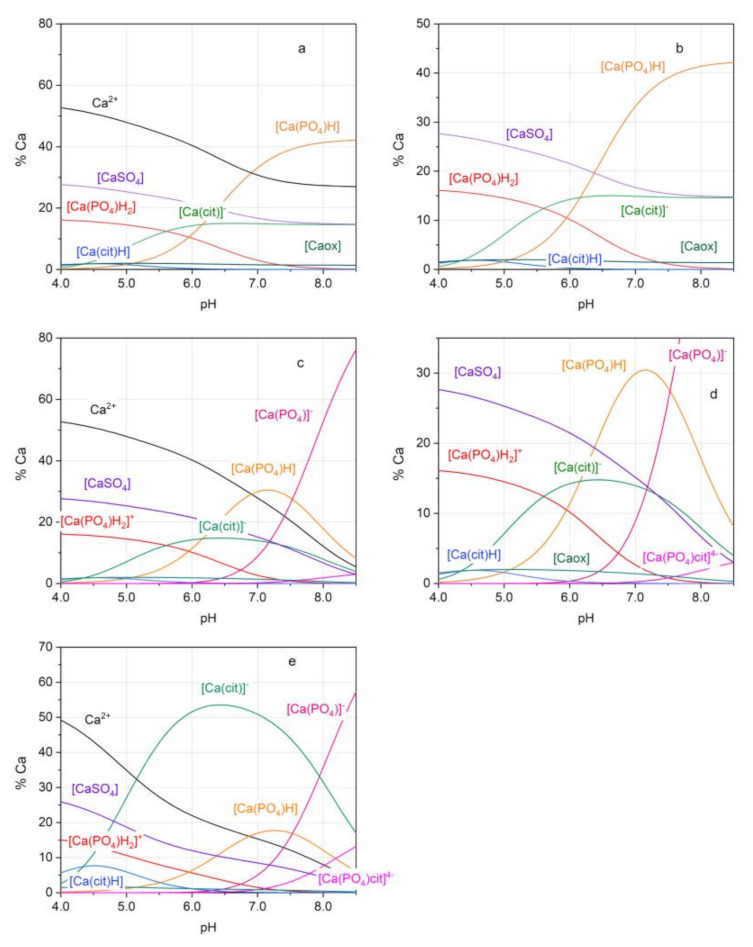
Speciation plots for the urine B obtained with Model 1: (**a**) full figure; (**b**) zoomed figure; and with Model 2: (**c**) full figure; (**d**) zoomed figure. (**e**) Speciation plots for urine B, obtained with Model 2 upon increasing total concentration of citrate to 4.0 × 10^−3^ mol L^−1^; cit = citrate anion; ox = oxalate anion.

**Table 1 molecules-26-03149-t001:** Overall (logβ) and partial (log*K*) formation constants at 25 and 37 °C, and at different ionic strengths, of the species in the water solutions containing Ca–phosphate and Ca–phosphate–citrate.

Species	Formation Constants						
	*I* = 0.16 mol L^−1^, 25 °C		*I* = 0.16 mol L^−1^, 37 °C		*I* = 0.1 mol L^−1^, 37 °C ^a^ [17]		Reference
	log*K*	logβ	log*K*	logβ	log*K*	logβ	
HPO_4_^2−^	11.79	11.79	11.64 ^b^	11.64	11.68	11.68	[18]
H_2_PO_4_^−^	6.84	18.63	6.83 ^b^	18.47	6.85	18.53	[18]
H_3_PO_4_	1.98	20.61	2.03 ^b^	20.51	2.04	20.57	[18]
Hcit^2− c^			5.80	5.80	5.83	5.83	[19]
H_2_cit^−^			4.31	10.11	4.33	10.16	[19]
H_3_cit			2.86	12.97	2.87	13.03	[19]
[KHPO_4_]^−^	0.50	12.29	0.58	12.22	0.62	12.30	[18]
[KPO_4_]^2−^	0.81	0.81	0.85	0.85	0.88	0.88	[18]
[Kcit]^2−^			0.56	0.56	0.59	0.59	[19]
[CaHcit]			2.03	7.83	2.05	7.88	[13]
[Cacit]^−^			3.49	3.49	3.56	3.56	[13]
			***I* = 0 mol L^−1^, 37 °C**		***I* = 0.1 mol L^−1^, 37 °C ^b^**		
[CaCl]^+^			−0.01	−0.01	0.02	0.02	[20]
[CaH_2_PO_4_]^+^					1.11	19.64	[14]
[CaHPO_4_]					2.02	13.70	[14]
[CaPO_4_]^−^					6.15 ± 0.02	6.15 ± 0.02	This work
[Ca(PO_4_)cit]^4− d^					8.3 ± 0.8	8.3 ± 0.8	This work

^a^ Values extrapolated from those at *I* = 0.16 M upon application of the Expanded Debye–Hückel equation [17]. ^b^ Values extrapolated from the data collected at different temperatures in [17]. ^c^ cit = citrate anion. ^d^ negligible species.

**Table 2 molecules-26-03149-t002:** Ionic strength (*I*_calc_) and saturation levels, estimated on five real urines. Model 1 does not consider the species [CaPO_4_]^−^ and [Ca(PO_4_)cit]^4−^. Model 2 considers the species [CaPO_4_]^−^ and [Ca(PO_4_)cit]^4−^.

Urine	pH ^1^		*I* _calc_		$\beta_{\mathrm{Caox}}{\;}^{2}$		$\beta_{{\mathrm{CaHPO}}_{4}}$	
		Model	1	2	1	2	1	2
**A**	7.50		0.107	0.107	2.57	2.25	1.34	1.15
**B**	7.24		0.148	0.148	6.76	5.74	5.71	4.73
**C**	7.06		0.064	0.064	2.52	2.33	3.42	3.10
**D**	6.84		0.148	0.146	4.67	4.52	2.37	2.27
**E**	6.57		0.103	0.103	9.60	9.49	2.93	2.87
**B* ^3^**	7.27		-	0.155	-	3.69	-	2.93

^1^ experimental value measured on the samples; ^2^ ox = oxalate anion; ^3^ B* = sample B with increased total citrate concentration up to 4.0 × 10^−3^ mol L^−1^.

## Data Availability

Data is contained within the article or Appendix A.

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
