# Peer review of "Critical Reappraisal of Methods for Measuring Urine Saturation with Calcium Salts"

_molecules, 2021, doi:10.3390/molecules26113149_

Round 1
Reviewer 1 Report
The manuscript by Silvia Berto et al. presents a piece of research dealing with the relevance of including two Ca2+complex species, namely [CaPO4]- and [CaPO4cit]4- (where cit = citrate), in the computer-based calculation models used to predict saturation in urine in order to evaluate the risk of forming kidney stones. The research, based on the potentiometric study of both chemical models and true urine samples, is well conducted and the conclusions, properly supported by the experimental results, are of clinical interest to the diagnosis and treatment of nephrolithiasis. The manuscript is well prepared and the language is proper and clear.
Then I recommend publication of this manuscript at its present state with the only suggestion of considering changing the word “Calculation” at page 1, line 15, to “Estimation”.
Author Response
Comments and Suggestions for Authors
The manuscript by Silvia Berto et al. presents a piece of research dealing with the relevance of including two Ca2+complex species, namely [CaPO4]- and [CaPO4cit]4- (where cit = citrate), in the computer-based calculation models used to predict saturation in urine in order to evaluate the risk of forming kidney stones. The research, based on the potentiometric study of both chemical models and true urine samples, is well conducted and the conclusions, properly supported by the experimental results, are of clinical interest to the diagnosis and treatment of nephrolithiasis. The manuscript is well prepared and the language is proper and clear.
Thanks to the reviewer for the kind comments.
Then I recommend publication of this manuscript at its present state with the only suggestion of considering changing the word “Calculation” at page 1, line 15, to “Estimation”.
Done
Reviewer 2 Report
The motivation to the work entitled ‘Critical reappraisal of methods for measuring urine saturation with calcium salts’ is the medical importance of the formation of kidney stones, which has a physicochemical background as it depends on the saturation, or supersaturation, of key components of urine. As expected for a biological solution highly dependent on individual metabolism, many factors play a role in the crystallization occurrence in the kidney. Both calculation and experimental work have been extensively performed in literature to improve knowledge on the role of components in inducing and inhibiting crystallization.
Here, the authors aimed at investigating the formation of ternary complexes (Ca-phosphate-citrate) which was accounted for discrepancies between JESS and EQUIL. Some required parameters were to be experimentally obtained using potentiometric titration and then real urine concentrations with and without the abovementioned ternary complex were compared regarding speciation diagrams and saturation.
Evidence shows that the ternary complex is not significantly formed thus should not be relevant for urine saturation calculations. Increased citrate concentrations have also not increased the formation such ternary complex. Results are interesting and relevant for the topic, but discussion can be improved as no further comments regarding the accuracy/discrepancy between models was given. The actual reappraisal of the methods was not fully given.
Introduction is clear and concise.
Results and Discussion
Figure 1: Quality must be improved. Both y and x-axis are blurred and there is a line on top of the figure that does not belong there.
Lines 91-93: ‘Concerning the dimeric species [Ca2H2(PO4)2]’, the authors claim to have already reported the non-formation of such species, which is corroborated by the experimental work. Yet, no reference is provided, and the relevance of this discussion to the paper, at this point, is underexplained.
Lines 95-108: The authors considered the technique as not sensitive enough and therefore results would not be used to define the formation constants of the calcium complexes. On the following paragraph, though, they say that the formation constants of parent complexes along with ‘all the data collected by titrating ternary mixtures containing Ca(II), citrate and phosphate’ were elaborated to get formation of ternary species. What does ‘elaborated’ means, exactly? Can the authors be more specific? If the experimental data was unsuitable, how was it used anyways? Or was it only partially used? Please explain.
Line 122: The word ‘doubtful’ does not seem adequate, given that the evidences lead to the conclusion that the formation of such complex is not to be expected.
Lines 139-140 should be merged with the previous paragraph.
Figure 2: The names of the complexes should be in the same colour of their respective lines, as is in Figure 3.
Line 150: Instead of saying ‘quite high pH values’, the reader would benefit from a more specific approach, for example, explicitly showing a range of pH.
Table 2S: Values for logK(CaPO4-) are different from the previously experimentally found in the work, and also referred to as ‘This work’. Where is the value used in Model 2 coming from?
Figure 3: While showing only the diagrams for one of the samples as to not overcrowd the paper with figures is understandable, a reason as to why sample B was specifically chosen should be given.
Including results for urine B with increased citrate concentrations in Table 2 should be considered for transparency.
Material and Methods
Line 222: ‘e.m.f’ appears as ‘e.f.m.’. Besides, this abbreviation needs to be described before using. The full term, electromotive force, only appers on line 237, not connected with the abbreviation.
Lines 225 – 228: Why does Ca2+:citrate was 1:1 but Ca2+:phosphate ranged between 1:1.5 and 1:4? Is there any particular reason?
Line 252: ‘drown’ should be ‘drawn’.
Information about the speciation model that is used in this work should be given in this section. There is an indication that Lithorisk was used, as it was developed within the group, however this is not directly mentioned anywhere. Authors should be specific as to which model was used and briefly mention how the model works, as to make the work more concise, transparent and reproducible.
Conclusions
Most text in the first paragraph does not belong to conclusions. It should be in the discussion section. Authors mentioned the discrepancies between literature models, JESS and EQUIL, as JESS introduced ternary complexed. They expect the same deviations with their group’s own model (LITHORISK), as it agrees with EQUIL. No further discussion was provided regarding these discrepancies. Due to the mention of these specific models here (and previous presentation of this scenario in the Introduction section), one would expect a comparison between models, or at least, a more detailed discussion on the results found and, based on these, if differences are expected between JESS and Lithorisk.
Line 263-264: The ‘aforementioned JESS’ was not (at least not directly) mentioned before in the paragraph. This makes unclear as to which of the previous statements refers to JESS.
Line 275-276: ‘uncommonly high urine pH’ is not specific. Furthermore, the formation of ternary complexes in terms of fraction of calcium is mentioned as, specifically, lower than 5 % in ‘uncommonly high urine pH’. In the discussion, it was always referred to as negligible or absent in normal pH, whilst at high pH, it was found up to 13 %. Please check.
Lines 278-285: Also belongs in the discussion.
Line 286-287: The authors mention to be ‘confident to conclude that the existence of this ternary complex is quite questionable’. This conclusion is contradictorily inconclusive. Saying it is questionable does not add any value to their own work. A suitable conclusion should be based on the evidence found in the work, which are far more conclusive than the word ‘questionable’ suggests.
References were not numbered and could not be properly checked.
Author Response
We thank the reviewer for the careful revision and for the useful suggestions. All the requests were satisfied or discussed. Here below the responses are reported point by point. The original reviewer’s comments were reported in black and the answers were reported in blue. The line numbers reported in the answers refer to the revised version of the manuscript. The revisions have been kept visible in the text.
Spell check was done.
Comments and Suggestions for Authors
The motivation to the work entitled ‘Critical reappraisal of methods for measuring urine saturation with calcium salts’ is the medical importance of the formation of kidney stones, which has a physicochemical background as it depends on the saturation, or supersaturation, of key components of urine. As expected for a biological solution highly dependent on individual metabolism, many factors play a role in the crystallization occurrence in the kidney. Both calculation and experimental work have been extensively performed in literature to improve knowledge on the role of components in inducing and inhibiting crystallization.
Here, the authors aimed at investigating the formation of ternary complexes (Ca-phosphate-citrate) which was accounted for discrepancies between JESS and EQUIL. Some required parameters were to be experimentally obtained using potentiometric titration and then real urine concentrations with and without the abovementioned ternary complex were compared regarding speciation diagrams and saturation.
Evidence shows that the ternary complex is not significantly formed thus should not be relevant for urine saturation calculations. Increased citrate concentrations have also not increased the formation such ternary complex. Results are interesting and relevant for the topic, but discussion can be improved as no further comments regarding the accuracy/discrepancy between models was given. The actual reappraisal of the methods was not fully given.
Introduction is clear and concise.
Results and Discussion
Figure 1: Quality must be improved. Both y and x-axis are blurred and there is a line on top of the figure that does not belong there.
The quality of Figure 1 was improved.
Lines 91-93: ‘Concerning the dimeric species [Ca2H2(PO4)2]’, the authors claim to have already reported the non-formation of such species, which is corroborated by the experimental work. Yet, no reference is provided, and the relevance of this discussion to the paper, at this point, is underexplained.
Lines 100-108: The reference was added and the point was better explained.
Lines 95-108: The authors considered the technique as not sensitive enough and therefore results would not be used to define the formation constants of the calcium complexes. On the following paragraph, though, they say that the formation constants of parent complexes along with ‘all the data collected by titrating ternary mixtures containing Ca(II), citrate and phosphate’ were elaborated to get formation of ternary species. What does ‘elaborated’ means, exactly? Can the authors be more specific? If the experimental data was unsuitable, how was it used anyways? Or was it only partially used? Please explain.
Lines 109-115 and 308-313: the text was modified to clarify the data analysis process.
Some information about the ISE-Ca data treatment were moved to the paragraph “Data treatment” where all the details about pH and pCa data analysis can be found.
The sentence “all the data collected by titrating ternary mixtures containing Ca(II), citrate and phosphate, were elaborated with the aim of evidencing the formation of ternary species.” was changed as “pH-metric data collected by titrating ternary mixtures containing Ca(II), citrate and phosphate, were analyzed with the aim of evidencing the formation of ternary species.” (lines 117-119) in order to clarify what type of data were used to optimize the formation constants.
Line 122: The word ‘doubtful’ does not seem adequate, given that the evidences lead to the conclusion that the formation of such complex is not to be expected.
Line 138: The word ‘doubtful’ was changed with “improbable”.
Lines 139-140 should be merged with the previous paragraph.
Lines 160-167: the text was not merged in order avoid dividing the Table 1 into two different pages; we hope that the readability could be anyway easy.
Figure 2: The names of the complexes should be in the same color of their respective lines, as is in Figure 3.
Figure 2: The color of the names of the complexes was modified.
Line 150: Instead of saying ‘quite high pH values’, the reader would benefit from a more specific approach, for example, explicitly showing a range of pH.
Line 173: the sentence was changed with “…5 natural urine samples having pH between 6.5 and 7.5”.
Table 2S: Values for logK(CaPO4-) are different from the previously experimentally found in the work, and also referred to as ‘This work’. Where is the value used in Model 2 coming from?
Table 2S: the value for logK(CaPO4-) is different from that reported in Table 1 because it refers at I = 0.16 M and it was extrapolated from that at I = 0.1 M upon application of an Extended Debye–Hückel equation. A note was added in order to clarify.
Figure 3: While showing only the diagrams for one of the samples as to not overcrowd the paper with figures is understandable, a reason as to why sample B was specifically chosen should be given.
Lines 197-198: the reasons as to why sample B was chosen were specified.
Including results for urine B with increased citrate concentrations in Table 2 should be considered for transparency.
Table 2: the results for urine B with increased citrate concentrations were inserted in Table 2.
Material and Methods
Line 222: ‘e.m.f’ appears as ‘e.f.m.’. Besides, this abbreviation needs to be described before using. The full term, electromotive force, only appers on line 237, not connected with the abbreviation.
Lines 263-264: ‘e.f.m.’ was changed with ‘e.m.f.’, the abbreviation was described and reported always as ‘e.m.f.’ in all the text.
Lines 225 – 228: Why does Ca2+:citrate was 1:1 but Ca2+:phosphate ranged between 1:1.5 and 1:4? Is there any particular reason?
Lines 272-274: the reason for the choice was clarified in the text.
Line 252: ‘drown’ should be ‘drawn’.
Line 327: done.
Information about the speciation model that is used in this work should be given in this section. There is an indication that Lithorisk was used, as it was developed within the group, however this is not directly mentioned anywhere. Authors should be specific as to which model was used and briefly mention how the model works, as to make the work more concise, transparent and reproducible.
In order to optimize the chemical model, we did not use Lithorisk, because this software was not devoted to optimize stability constants. Lithorisk provides the saturation state of urine samples based on the urinary parameters (those reported in Table 1S), in a report easy to use for inexpert people. More detailed information about Lithorisk can be found in the references.
The model optimization was done by the BSTAC software as reported in the paragraph (line 302). The model so optimized will be inserted in Lithorisk to improve the saturation level estimation and for better manage nephrolithiasis. The description of how the BSTAC software works was previously reported in the text (lines 302-308) and can be find in the references. Moreover, in order to verify how model variations affect the concentration of the single species present in urine, the species distribution diagrams were drawn by HySS, as reported in the text (line 327).
To clarify how the results reported in this work can be used to improve the estimation of the overall risk of kidney stones by Lithorisk software, a sentence was added at the end of the conclusion paragraph (lines 366-369).
Conclusions
Most text in the first paragraph does not belong to conclusions. It should be in the discussion section. Authors mentioned the discrepancies between literature models, JESS and EQUIL, as JESS introduced ternary complexed. They expect the same deviations with their group’s own model (LITHORISK), as it agrees with EQUIL. No further discussion was provided regarding these discrepancies. Due to the mention of these specific models here (and previous presentation of this scenario in the Introduction section), one would expect a comparison between models, or at least, a more detailed discussion on the results found and, based on these, if differences are expected between JESS and Lithorisk.
A discussion paragraph was inserted in the manuscript and a part of the text, before included in the Conclusions, was shifted there, as suggested.
The aim of this work was to define whether the species inserted in JESS, but so far neglected in Lithorisk, could solve the discrepancies between the results of the two software. The experimental results obtained reveal that the species neglected by the Lithorisk model are not relevant in the chemical system. Anyway, during the investigation, the necessity to define the formation constant for the parent species [CaPO4]- lead us to consider it and to observe that it must be inserted in the model. These observations lead to consider unnecessary a direct comparison between the results obtained by JESS and Lithorisk because the divergences on the chemical models used were not solved.
Lines 263-264: The ‘aforementioned JESS’ was not (at least not directly) mentioned before in the paragraph. This makes unclear as to which of the previous statements refers to JESS.
The sentence was removed during the revision process.
Lines 275-276: ‘uncommonly high urine pH’ is not specific. Furthermore, the formation of ternary complexes in terms of fraction of calcium is mentioned as, specifically, lower than 5 % in ‘uncommonly high urine pH’. In the discussion, it was always referred to as negligible or absent in normal pH, whilst at high pH, it was found up to 13 %. Please check.
Line 352: the pH of urine was specified in coherence with the statements reported before.
Lines 278-285: Also belongs in the discussion.
See the comment reported before (new Discussion section).
Line 286-287: The authors mention to be ‘confident to conclude that the existence of this ternary complex is quite questionable’. This conclusion is contradictorily inconclusive. Saying it is questionable does not add any value to their own work. A suitable conclusion should be based on the evidence found in the work, which are far more conclusive than the word ‘questionable’ suggests.
Lines 363-364: the sentence was changed with “The overall results of the present study make us confident to conclude that the existence of the ternary complex can be rejected.”.
References were not numbered and could not be properly checked.
Sorry, in the original version the numbers were present, probably there was a problem during the upload of the manuscript. We will check more carefully the upload of the revised version.

Reviewer 3 Report
Title: Critical reappraisal of methods for measuring urine saturation with calcium salts
Abstract
Background: metabolic and physicochemical evaluation is recommended to manage patients with nephrolithiasis. Calculation of the state of saturation (β values) is often included in diagnostic work-up and is preferably performed by calculations. Free concentrations of constituent ions are estimated by considering the main ionic soluble complexes. It is contended that this approach is liable to overestimation of β values because some complexes may be overlooked. A recent report found that β values could be significantly lowered upon addition a new so far neglected complexes, [Ca(PO4)Cit]4- and [Ca2H2(PO4)2].
Methods: to assess whether these complexes can form in urine, the water systems Ca-phosphate-citrate was investigated by potentiometric titrations. The stability constants of the parent binary complexes [Cacit]- and [CaPO4]-, and the coordination tendency of the ligand PO43- towards [Ca(cit)]- to form the ternary complex, were estimated. βCaOx and βCaHPO4 were then calculated on 5 natural urines by chemical models including or not the species [CaPO4]- and [Ca(PO4)cit]4-.
Results: species distribution diagrams show that the species [Ca(PO4)cit]4- was only noticeable at pH>8.5 and below 10% of total calcium. β values estimated on natural urine resulted slightly lowered by the formation of [CaPO4]- species whereas [Ca(PO4)cit]4- resulted irrelevant. Conclusions: while [CaPO4]- species may have some impact on saturation levels at higher pHs, the existence of ternary complex is quite questionable.
The conclusion is rather vague and leave the reader without proper take home message.
The method reads like a Aim/Objective , please amend accordingly
The Method, Results and Discussion Sections are well organized and written. I have no major issue.
Conclusion: The current Conclusion needs to be improved by trimming the duplication sentences
Author Response
Comments and Suggestions for Authors
Title: Critical reappraisal of methods for measuring urine saturation with calcium salts
Abstract
Background: metabolic and physicochemical evaluation is recommended to manage patients with nephrolithiasis. Calculation of the state of saturation (β values) is often included in diagnostic work-up and is preferably performed by calculations. Free concentrations of constituent ions are estimated by considering the main ionic soluble complexes. It is contended that this approach is liable to overestimation of β values because some complexes may be overlooked. A recent report found that β values could be significantly lowered upon addition a new so far neglected complexes, [Ca(PO4)Cit]4- and [Ca2H2(PO4)2].
Methods: to assess whether these complexes can form in urine, the water systems Ca-phosphate-citrate was investigated by potentiometric titrations. The stability constants of the parent binary complexes [Cacit]- and [CaPO4]-, and the coordination tendency of the ligand PO43- towards [Ca(cit)]- to form the ternary complex, were estimated. βCaOx and βCaHPO4 were then calculated on 5 natural urines by chemical models including or not the species [CaPO4]- and [Ca(PO4)cit]4-.
Results: species distribution diagrams show that the species [Ca(PO4)cit]4- was only noticeable at pH>8.5 and below 10% of total calcium. β values estimated on natural urine resulted slightly lowered by the formation of [CaPO4]- species whereas [Ca(PO4)cit]4- resulted irrelevant. Conclusions: while [CaPO4]- species may have some impact on saturation levels at higher pHs, the existence of ternary complex is quite questionable.
The conclusion is rather vague and leave the reader without proper take home message.
The method reads like a Aim/Objective , please amend accordingly
We thank the reviewer for the useful suggestions.
Some sentences of the Abstract were changed in order to improve the readability. The aim of the work was shifted in the Background section and the Conclusions were declared more sharply.
Spell check was done.
The Method, Results and Discussion Sections are well organized and written. I have no major issue.
Conclusion: The current Conclusion needs to be improved by trimming the duplication sentences.
A discussion paragraph was inserted in the manuscript and a part of the Conclusions was shifted there. The Conclusions paragraph is now neater and some concepts were better defined to better convey the message of the work.
Round 2
Reviewer 2 Report
Manuscript ‘Critical reappraisal of methods for measuring urine saturation with calcium salts’ was revised.
All the comments were satisfactorily addressed. I believe clarity and readability was significantly improved.